# Language Model Quality Correlates with Psychometric Predictive Power in Multiple Languages

**Ethan Gotlieb Wilcox**💯  **Clara Meister**💯  **Ryan Cotterell**💯  **Tiago Pimentel**💪,💯

💯ETH Zürich  💪University of Cambridge

{`ethan.wilcox`, `clara.meister`, `ryan.cotterell`, `tiago.pimentel`}@inf.ethz.ch

## Abstract

Surprisal theory (Hale, 2001; Levy, 2008) posits that a word's reading time is proportional to its surprisal (i.e., to its negative log probability given the proceeding context). It has been empirically tested using surprisal estimates from language models (LMs). Under the premise that surprisal theory holds, we would expect that higher quality language models, whose predictions are more accurate, provide more powerful predictors of human reading behavior—a conjecture we dub the **quality–power (QP) hypothesis**. Unfortunately, empirical support for the QP hypothesis is mixed. Some studies in English have found correlations between LM quality and psychometric predictive power, but other studies using Japanese data, as well as using larger English LMs, find no such correlations. In this work, we conduct a systematic crosslinguistic assessment of the QP hypothesis. We train LMs from scratch on small- and medium-sized datasets from 13 languages (across five language families) and assess their ability to predict eye tracking data. We find correlations between LM quality and psychometric predictive power in eleven of these thirteen languages, suggesting that, within the range of model classes and sizes tested, better language models provide better predictors of human language processing behaviors.

⊙ https://github.com/rycolab/quality-power-hypothesis

## 1 Introduction

The relationship between a word's predictability and its reading time (RT) is an important object of study in psycholinguistics because it allows us to draw insights about how humans process sentences (Smith and Levy, 2013; Kuperberg and Jaeger, 2016; Shain et al., 2022). Extensive prior research—across many datasets, populations, reading-time measurement methodologies and even languages—has found that the more predictable a word is, the faster it will be to process (Kuribayashi et al., 2021; Meister et al., 2021; Smith and Levy, 2013; Wilcox et al., 2020, 2023; Shain et al., 2022; de Varda and Marelli, 2022, 2023). Beyond these empirical findings, the relationship between predictability and reading time has received formal treatment, forming the basis of **surprisal theory** (Hale, 2001; Levy, 2008). In surprisal theory, a word's predictability is operationalized in terms of its **surprisal**, or contextual negative log-likelihood: $s_✓(w_t \mid \boldsymbol{w}_{<t}) = -\log p_✓(w_t \mid \boldsymbol{w}_{<t})$.[1] In practice, however, we do not know the contextual probability of words according to the data-generating distribution $p_✓$. Thus, it is common to estimate surprisal using the distribution over words from a language model $p_\theta$ instead, i.e., $s_\theta(w_t \mid \boldsymbol{w}_{<t}) = -\log p_\theta(w_t \mid \boldsymbol{w}_{<t})$.

Recent advances in language modeling have drastically increased the quality of surprisal estimates from language models (Hoffmann et al., 2022; Chowdhery et al., 2022; OpenAI, 2023). The topic of interest of this work is whether this increased quality also leads to an increase in a language model's **psychometric predictive power** (Frank and Bod, 2011; Goodkind and Bicknell, 2018), i.e., how well its surprisal estimates can be used to predict RTs. To put it simply, we ask: Are better language models more powerful predictors of human behavior? If surprisal theory holds, we would expect this power to correlate positively with language models' quality. We will refer to this conjecture as the **quality–power hypothesis**, abbreviated as the QP hypothesis.

Empirical evidence for the QP hypothesis is mixed. When tested on English corpora, some studies have found that models that place higher probability on held-out data do tend to be better at predicting the pattern of human reading times (Goodkind and Bicknell, 2018; Wilcox et al., 2020). More recent work, however, has discovered two em-

---

[1]Empirical results mostly support the existence of a linear relationship between surprisal and reading times (Smith and Levy, 2013; Shain et al., 2022; Wilcox et al., 2023): They find that surprisal is a better psychometric predictor than raw probabilities, and that non-linear functions of surprisal are not better than linear ones.

pirical exceptions: The first empirical shortcoming was presented in Kuribayashi et al. (2021)—the only[2] non-English study (to the best of our knowledge) investigating the quality–power relationship.[3] Specifically, Kuribayashi et al. find that the QP hypothesis does not hold in Japanese. The second shortcoming was identified by Shain et al. (2022) and Oh and Schuler (2023), who observe that the quality–power relationship does not hold for the *best* (and most recent) language models, suggesting state-of-the-art language models might not be aligned with human incremental predictions.

In this work we present two empirical studies that investigate these shortcomings. First, we conduct a systematic crosslinguistic analysis of the QP hypothesis, using small- and medium-sized LMs.[4] We train 104 language models from scratch on data from 13 languages, using subsamples of the Wiki40b dataset (Guo et al., 2020). Within each language, we operationalize LMs' qualities as their average per-word cross entropy on a test set. We then use the surprisals under these models to predict human reading times on a multilingual corpus of eye tracking data. We find that, in eleven out of thirteen languages, there is a significant correlation between a language model's quality and its surprisals' psychometric predictive power.

To investigate the second shortcoming, we note that a (potential) issue with the findings presented in Oh and Schuler (2023) and Shain et al. (2022) is that the training data for these models is not publicly available, and might plausibly includes the reading time corpora against which their psychometric predictive power was evaluated. We discuss why this might present an issue, and perform an investigation of the relationship between data leakage and model's predictive power in §5, albeit with null results.

## 2 Surprisal vs Reading Times

Originally proposed by Hale (2001), **surprisal theory** posits that the amount of effort (and there-

---

[2]Concurrent to this study, de Varda and Marelli (2023) investigate the QP hypothesis in a crosslinguistic setup. Unlike our work, they rely on pre-trained multilingual models whereas we make use of in-house trained monolingual models. This allows them to investigate larger models than us, but also it introduces biases inherent to the use of multilingual models.

[3]See Blasi et al. (2022) for a survey on the importance of testing cognitive hypotheses beyond English.

[4]Note that our best in-house trained language models are still of far lower quality than the large language models analyzed by Shain et al. (2022) and Oh and Schuler (2023).

fore time) a reader must spend to process a word is a monotonically increasing function of its surprisal, where surprisal is a measure of a word's information content (Shannon, 1948; Cover and Thomas, 2006). Surprisal theory has attracted lots of attention over the years, by both the natural language processing and cognitive science communities (Hale, 2001, 2003, 2016; Keller, 2004; van Schijndel and Schuler, 2016; van Schijndel and Linzen, 2018; Shain, 2019, 2021; Shain and Schuler, 2021, 2022; Wilcox et al., 2020; Meister et al., 2021, 2022; Hoover et al., 2022; Oh et al., 2021; Oh and Schuler, 2022, 2023; Kuribayashi et al., 2021, 2022, *inter alia*).

One way surprisal theory has been evaluated is by quantifying surprisal's predictive power. If a word's surprisal influences reading times, we should see evidence in naturalistic reading data. Psycholinguists thus train regressors $p_\phi$ to predict reading times: $r(w_t) \sim p_\phi(r \mid \cdot)$, where $p_\phi$'s input is defined as a vector of baseline features $\mathbf{x}_t$, comparing the regressors' performances when surprisal is included and when it is not. Specifically, the predictive power of surprisal is quantified as the difference in the log-likelihood (llh) of reading time data under the two different regressors:

$$\Delta_{\text{llh}} = \text{llh}(p_\phi(r \mid \mathbf{x}_t, s_\checkmark)) - \text{llh}(p_\phi(r \mid \mathbf{x}_t)) \quad (1)$$

Note that this is an ideal $\Delta_{\text{llh}}$, which assumes we have access to surprisals from the data-generating distribution, which we write as $s_\checkmark$ (as a shorthand for $s_\checkmark(w_t \mid \boldsymbol{w}_{<t})$).

### 2.1 Estimated Surprisal

As noted in §1, we do not have access to $p_\checkmark$ and thus, we cannot compute $s_\checkmark(w_t \mid \boldsymbol{w}_{<t})$ directly. We must therefore estimate it, which we do with the use of a language model $p_\theta$. Formally, the language models $p_\theta$ that we discuss here are autoregressive probability distributions over $\overline{\mathcal{V}} \stackrel{\text{def}}{=} \mathcal{V} \cup \{\text{EOS}\}$, defined as the conditional distribution $p_\theta(\cdot \mid \boldsymbol{w}_{<t})$.

A language model's parameters are typically chosen with the objective of minimizing its cross entropy with the true distribution:[5]

$$\text{H}_{\checkmark \to \theta}(\boldsymbol{W}) = \sum_{\boldsymbol{w} \in \mathcal{S}} p_\checkmark(\boldsymbol{w}) \sum_{t=1}^{|\boldsymbol{w}|} s_\theta(w_t \mid \boldsymbol{w}_{<t}) \quad (2)$$

---

[5]In practice, as we do not have access to $p_\checkmark$, this is done using a Monte Carlo estimator and a training dataset $\mathcal{D}_{\text{trn}} = \{\boldsymbol{w}^{(n)}\}_{n=1}^N$, where $\boldsymbol{w}^{(n)}$ are assumed to be sampled i.i.d. from $p_\checkmark$: $\text{H}_{\checkmark \to \theta}(\boldsymbol{W}) \approx \frac{1}{N} \sum_{n=1}^N \sum_{t=1}^{|\boldsymbol{w}^{(n)}|} s_\theta(w_t^{(n)} \mid \boldsymbol{w}_{<t}^{(n)})$

where $\mathcal{S} \stackrel{\text{def}}{=} \mathcal{V}^* \circ \{\text{EOS}\}$. A language model's training, then, pushes it towards being a better surprisal estimator. This can be shown clearly when we rewrite the cross entropy as the sum of $p_{\checkmark}$'s entropy and the Kullback–Leibler (KL) divergence between $p_{\checkmark}$ and $p_{\theta}$:

$$\text{H}_{\checkmark \to \theta}(\boldsymbol{W}) = \text{H}_{\checkmark}(\boldsymbol{W}) + \text{KL}(p_{\checkmark} \mid\mid p_{\theta}) \quad (3)$$

where the KL measures a statistical "distance" between two distributions, being zero only when $p_{\theta} = p_{\checkmark}$. Importantly, language models are not perfect estimates of $p_{\checkmark}$, and so they do not achieve this minimum. Intuitively, $\text{H}_{\checkmark \to \theta}(\boldsymbol{W})$ should then give us a sense for how similar $p_{\theta}$ is to $p_{\checkmark}$, and, thus, how similar $s_{\theta}$ is to $s_{\checkmark}$. These surprisal estimates $s_{\theta}$ are then in turn used to estimate surprisal's predictive power

$$\widehat{\Delta}_{\text{llh}} = \text{llh}(p_{\phi}(r \mid \mathbf{x}_t, s_{\theta})) - \text{llh}(p_{\phi}(r \mid \mathbf{x}_t)) \quad (4)$$

We can now state the QP hypothesis in terms of these more formal definitions: the QP hypothesis predicts that as the cross entropy $\text{H}_{\checkmark \to \theta}(\boldsymbol{W})$ decreases, a model's surprisal values should become better predictors of reading times, leading to larger $\widehat{\Delta}_{\text{llh}}$.

## 3 Experimental Setup

**Measuring Predictive Power.** As discussed in §2, and following previous work in this area (Goodkind and Bicknell, 2018; Wilcox et al., 2020), we quantify a language model's predictive power as its delta log likelihood ($\widehat{\Delta}_{\text{llh}}$). When predicting the reading time of word $w_t$ in context, we will use features associated with $w_t$ and its two preceding words $w_{t-1}$, and $w_{t-2}$ to account for spillover effects.[6] We will refer to this combined set of three words as our **target words**. The vector of variables in our baseline regressor $\mathbf{x}_t$ include the log unigram frequency and the length (in characters) of our target words. The variables for our comparison regressor include $\mathbf{x}_t$ plus two additional variables: the surprisal and the Rényi entropy of our target words. Rényi entropy is a generalization of Shannon entropy which measures the expected surprisal of a word, given its context, and has been shown previously to impact reading times above and beyond surprisal (Pimentel et al., 2023). Results from regressors that use just surprisal, or

just Rényi entropy as additional predictors are presented in App. A and B. To quantify the power of the language model, we report $\widehat{\Delta}_{\text{llh}}$ (as laid out in eq. (4)) measured across ten folds of held-out data. A positive $\widehat{\Delta}_{\text{llh}}$ means that including surprisal and entropy as predictors increases predictive power over reading times. A negative value of $\widehat{\Delta}_{\text{llh}}$ is also possible and it implies overfitting.

**Eye Tracking Data.** We measure $\widehat{\Delta}_{\text{llh}}$ on MECO (Siegelman et al., 2022), a multilingual corpus of eye tracking data collected on simple Wikipedia-style articles from thirteen different languages. Breaking the languages into their respective families, we have eight Indo-European languages (Dutch, English, German, Greek, Italian, Norwegian, Russian and Spanish), two Uralic languages (Finnish, Estonian), one Afro-Asiatic language (Hebrew), one Koreanic language (Korean) and one Turkic language (Turkish). Articles in the MECO dataset undergo a multi-step translation process to ensure that meaning was preserved across languages. Following previous studies (Smith and Levy, 2013; Wilcox et al., 2020), we use each word's gaze duration as our measure of reading time.[7] For dependent variables in our regressors, we use cross-participant averages, and treat skipped words as having a reading time of zero (as previously done by Rayner et al., 2011; Pimentel et al., 2023; Wilcox et al., 2023).

**Language Models.** We train language models on different language sections of the Wiki40B dataset (Guo et al., 2020). The data for each language is pre-split into a training, validation and test set in Wiki40B. Before training, we fit a UnigramLM tokenizer with a vocabulary size of 32k to each language's entire training set. This tokenizer is then shared across all LMs within that language. We then subsample each language's training set into sets with 1M, 3M, 10M, 30M, 100M, 300M and 1B tokens. We train models independently on each of this subsets, as well as on the entire training set available for that language. All models are trained using fairseq (Ott et al., 2019), following their recommended language modeling hyper-parameters. We use a standard decoder-only transformer with 6 layers, a context window size of 512 tokens, and shared input–output embeddings. We train our models using Adam (Kingma and Ba, 2015), with

---

[6] Spillover effects, common in reading, are when properties of a word impact the reading times of subsequent words.

[7] Gaze duration is the amount of time between a readers first fixation on a word and the first time their gaze leaves the word.

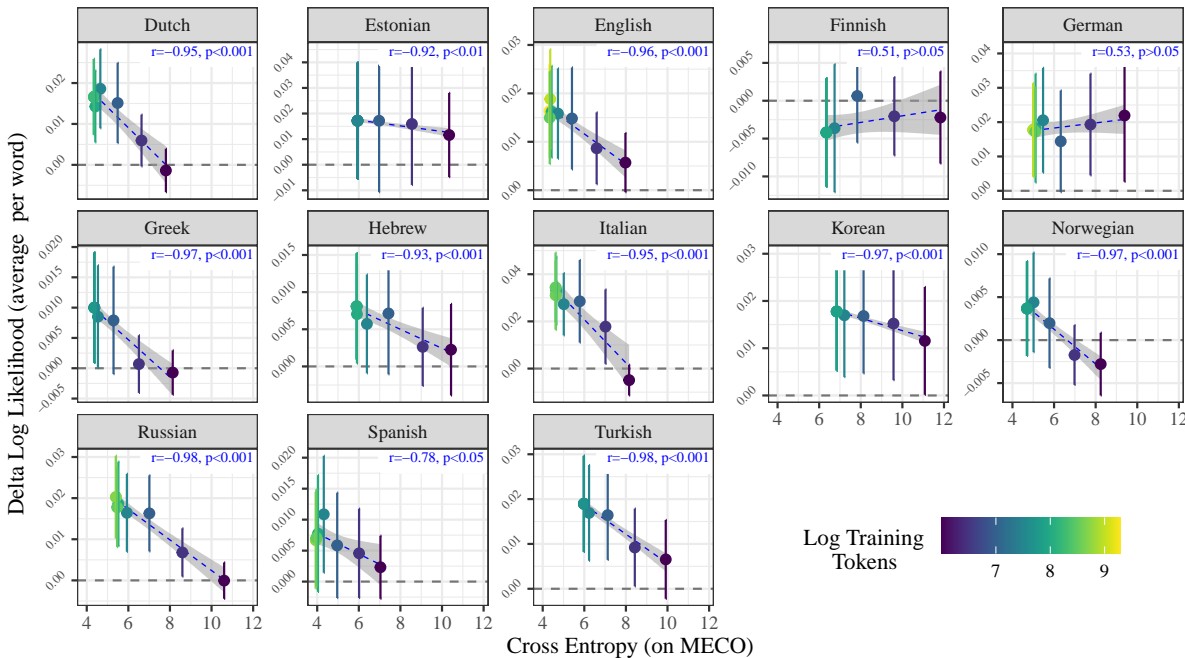

Figure 1: **Results.** Error bars are 95% confidence intervals on heldout data. Log training tokens is given in base 10. Fits are linear lines of best-fit and 95% confidence intervals. Blue labels show the $r$- and $p$-value of a Pearson correlation test between cross entropy and delta log-likelihood. We find a negative correlation in 11 out of 13 languages tested.

a learning rate of $5e^{-4}$, 4000 warm-up updates, and dropout of 0.1. We evaluate our models after each full epoch on their respective validation sets, using early stopping with a patience of 3 epochs. As our measure of language model quality, we report their cross entropy on the MECO dataset.

## 4 Results of the Crosslinguistic Analysis

The results for our crosslinguistic analysis can be seen in Fig. 1, with the languages in the different facets, $H_{\checkmark \to \theta}$ on the $x$-axis and $\widehat{\Delta}_{\text{llh}}$ on the $y$-axis. First, we note that the majority of language models are able to achieve $\widehat{\Delta}_{\text{llh}}$ above zero, indicating that surprisal and entropy values from the models are helpful in predicting reading times regardless of their size. This is in line with previous work that has found that surprisal and entropy have psychometric predictive power. That being said, the surprisal and entropy from our worse models (which were trained on only 1 million words) typically do not lead to positive $\widehat{\Delta}_{\text{llh}}$, which is not unexpected from models that are such poor estimates of $p_{\checkmark}$ (as evinced by their large cross entropy).

Turning to the relationship between cross entropy and $\widehat{\Delta}_{\text{llh}}$, visually, it is quite clear that the majority of languages exhibit a positive relationship between language model quality and psychometric predictive power, i.e., a negative relationship between $\widehat{\Delta}_{\text{llh}}$ and cross entropy. To

test this trend statistically, we fit linear regression models with $\widehat{\Delta}_{\text{llh}}$ as the sole dependent variable and cross entropy as a predictor. We find a significant effect of cross entropy on $\widehat{\Delta}_{\text{llh}}$ in 11 out of our 13 analyzed languages.[8] We do not find a significant effect of cross entropy in Finnish or German. Results are shown in Fig. 1, where we also report the Pearson correlation coefficient between the cross entropy and mean $\widehat{\Delta}_{\text{llh}}$ of each language model. The high value of $r$ quantitatively confirms the visual trends seen in Fig. 1.

Finally, we also run a regression model on all our data, pooled across languages. Here, in addition to the fixed effect of $H_{\checkmark \to \theta}$, we include random slopes and intercepts for each language.[9] In this regression we find a significant effect of $H_{\checkmark \to \theta}$ on $\widehat{\Delta}_{\text{llh}}$ ($\beta = -0.0026, p < 0.01$). Together, these results indicate that across a range of languages and model sizes, there is a consistent relationship between a language model's quality and its predictive power.

---

[8]Dutch ($\beta = -0.0047$, $p < 0.001$), English ($\beta = -0.0021$, $p < 0.001$), Greek ($\beta = -0.0028$, $p < 0.001$), Hebrew ($\beta = -0.0013$, $p < 0.001$), Italian ($\beta = -0.0088$, $p < 0.001$), Norwegian ($\beta = -0.0020$, $p < 0.001$), Russian ($\beta = -0.0037$, $p < 0.001$), Turkish ($\beta = -0.0031$, $p < 0.001$), Spanish ($\beta = -0.0016$, $p < 0.01$), Estonian ($\beta = -0.0008$, $p < 0.05$) and Korean ($\beta = -0.0009, p < 0.05$).

[9]Our lmer call was dll $\sim$ x-ent + (x-ent | lang)

## 5   Analysis of Data Leakage

As mentioned in the introduction, prior studies have found that the relationship between LM quality and psychometric predictive power does not hold for large, contemporary LMs (Oh and Schuler, 2023; Shain et al., 2022). One potential concern for both of these studies is that they used models whose training data is not publicly available, and is so large in scale, that it quite feasibly suffers from data leakage. That is, as human reading time datasets were publicly available, models might have been trained and tested on the same materials.

**Motivation.** There are a number of ways in which a model's surprisal estimates $s_\theta$ for leaked data could provide a poorer fit to reading times. Importantly, models are likely to underestimate surprisal on training data, and there could feasibly be a difference in the degree to which high surprisal vs. low surprisal values are underestimated. Such a difference could hurt the ability to use surprisal to predict RTs when restricted to linear regressors, which are the main focus of many RT analyses since the relationship between RT and surprisal has been observed to be linear in nature (Smith and Levy, 2008, 2013; Shain et al., 2022). Explicitly, since the aforementioned underestimation might disproportionately skew a subset of surprisal estimates, linear regressors would be unable to model this altered relationship.

**Experimental Setup.** To assess whether leakage could be a confounding factors in previous evidence against the QP hypothesis, we train models with different amounts of leaked data and investigate its impact on $\widehat{\Delta}_{\text{llh}}$. To do so, we create versions of our full (i.e., not subsampled) training datasets in each language that include 50% or 100% of the MECO materials for that language. We then evaluate these models on MECO as before.

**Results.** Results for this experiment can be seen in Fig. 2, which shows the average $\widehat{\Delta}_{\text{llh}}$ across languages. The average $\widehat{\Delta}_{\text{llh}}$ for each amount of leakage is also printed above the bars. Although the version of MECO with 0% leaked data does achieve the highest $\widehat{\Delta}_{\text{llh}}$, the confidence intervals overlap with each other. To confirm this statistically, we first compute delta log likelihood estimated for each word–RT pair in our corpus:

$$\widehat{\delta}_{\text{llh}} = \log p_\phi(r \mid \mathbf{x}_t, s_\theta)) - \log p_\phi(r \mid \mathbf{x}_t) \quad (5)$$

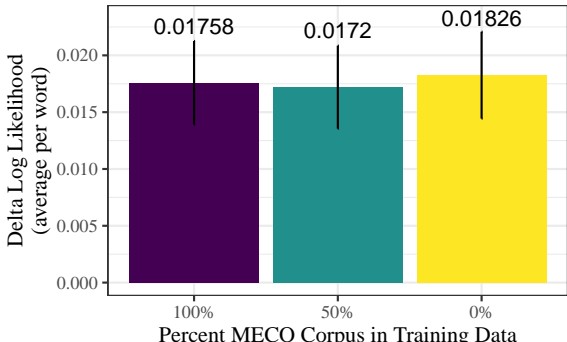

Figure 2: **Results of the Data Leakage Experiment**: Error bars are 95% confidence intervals averaged across languages. Average values of $\widehat{\Delta}_{\text{llh}}$ in each category are printed above the bars.

and then fit a mixed-effects regression model with these $\widehat{\delta}_{\text{llh}}$ as the response variable and a single categorical fixed effect indicating whether the $\widehat{\delta}_{\text{llh}}$ was derived from a data-leaked model or not. We run separate tests comparing between 0% and 50% and comparing between 0% and 100% leakage. We included random slopes and intercepts for each language. The degree of data leakage was not a significant predictor of $\widehat{\delta}_{\text{llh}}$ ($p > 0.05$). Although our results can not be taken to show, definitively, that the breakdown in the QP relationship observed by Shain et al. (2022) and Oh and Schuler (2023) is or is not due to data leakage, they suggest that this is likely not the primary cause. We thus encourage researchers to explore other hypotheses about why these larger models appear to be poorer predictors of human reading behavior.

## 6   Conclusion

This paper investigates the QP hypothesis, i.e., the relationship between a language model's quality and its psychometric predictive power. It looks specifically at two settings in which there is some evidence against the hypotheses by providing (i) a crosslinguistic assessment of its predictions and (ii) an investigation of the role of data leakage in model predictive power. Although our results demonstrate that, at small- and medium-sized data scales, the QP hypothesis holds in the large majority of languages tested, our results do raise some questions. Perhaps the most pressing of these has to do with our results for Finnish and German, which showed *negative* (albeit non-significant) correlations between quality and power. Further testing in these two languages, thus, is an important next step for future research.

## Limitations

One empirical limitation of this study is that we did not assess the QP hypothesis in Japanese (which is not in the MECO dataset). Negative results in that language were one of the primary motivations for this work. Assuming that the negative Japanese results still hold, what should we make of this? One possibility is that the results from Japanese might be due to its writing system, which combines syllabaries (the kana) and logosyllabic characters (the kanji). All of the scripts investigated here either use alphabets or, in the case of Korean, blends between alphabets and syllabaries. Our results strongly suggest that the QP hypothesis does hold across languages that share a common approach to writing. A next logical step would be to test this hypothesis across writing systems in a more systematic way.

## Ethics Statement

The MECO dataset required human participation. We refer the reader to the associated publication for their ethical guidelines in subject interactions. We do not foresee any ethical issues with our work.

## Acknowledgements

EGW would like to acknowledge support from an ETH Postdoctoral Fellowship. CM is funded by a Google PhD Fellowship. TP is funded by a Facebook PhD Fellowship.

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

## A  Results with Surprisal

In order to facilitate a closer comparison with previous analyses, which have not looked at the effect of entropy (e.g., Goodkind and Bicknell, 2018), we conduct a version of our analysis with surprisal as the sole non-baseline predictor. Results are presented in Fig. 3. As with the results in the main section, we find a negative relationship between $\widehat{\Delta}_{\text{llh}}$ and $\text{H}_{\checkmark \to \theta}$ across languages. To test this trend statistically, we fit linear regression models with $\widehat{\Delta}_{\text{llh}}$ as the response variable and $\text{H}_{\checkmark \to \theta}$ as the sole predictor variable. We find a significant effect of $\text{H}_{\checkmark \to \theta}$ on $\widehat{\Delta}_{\text{llh}}$ in Dutch, Estonian, Greek, Italian, Norwegian, Russian, Turkish ($p < 0.001$), English, Hebrew and Spanish ($p < 0.05$), but not in Finnish or Korean. In German, we find a positive effect of $\widehat{\Delta}_{\text{llh}}$ on $\text{H}_{\checkmark \to \theta}$ ($p < 0.05$). As before, we also run a mixed-effects regression model that tests the effect cross-linguistically with random intercepts and slopes for language and log-number of training tokens. We find a significant effect of $\widehat{\Delta}_{\text{llh}}$ on $\text{H}_{\checkmark \to \theta}$ ($p < 0.05$). Overall, these results are consistent with the results in the main body of the paper.

## B  Results with Entropy

Additionally, we conduct a version of our analysis that includes only (Rényi) entropy as a non-baseline predictor. Results are presented in Fig. 4. Again, we find that the majority of languages show a negative relationship between $\text{H}_{\checkmark \to \theta}$ and $\widehat{\Delta}_{\text{llh}}$. Conducting the same statistical analysis as discussed in the main body of the paper, we find a significant effect of $\widehat{\Delta}_{\text{llh}}$ on $\text{H}_{\checkmark \to \theta}$ in Dutch, Korean, Russian, Turkish ($p < 0.001$), English, German, Greek, Italian ($p < 0.01$), Norwegian, Hebrew and Spanish ($p < 0.05$), but no effect in Finnish. We find a positive effect in Estonian ($p < 0.001$). We also run a mixed-effects regression model that tests the effect cross-linguistically with random intercepts and slopes for language and log-number of training tokens. We find a significant effect of $\widehat{\Delta}_{\text{llh}}$ on $\text{H}_{\checkmark \to \theta}$ ($p < 0.05$).

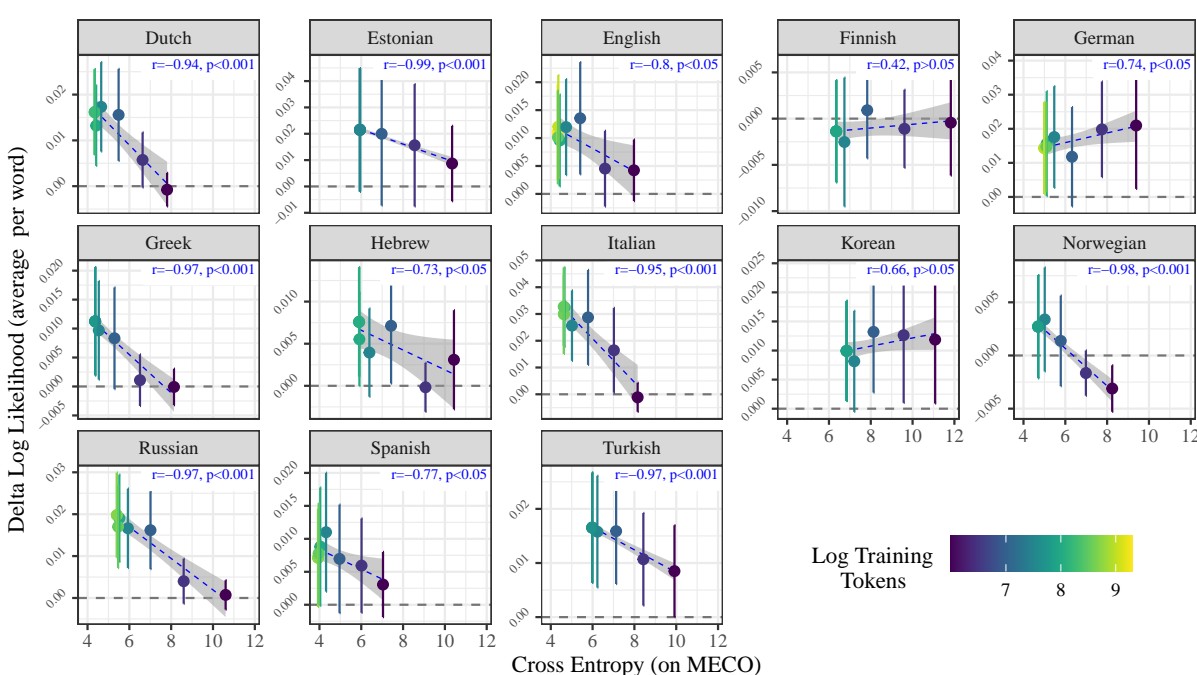

Figure 3: **Results with Surprisal Only**: Error bars are 95% confidence intervals on heldout data. Fits are linear lines of best-fit and 95% confidence intervals. Blue labels show the $r$- and $p$-value of a Pearson correlation test between cross entropy and delta log-likelihood. We find a negative correlation in 10 out of 13 languages tested.

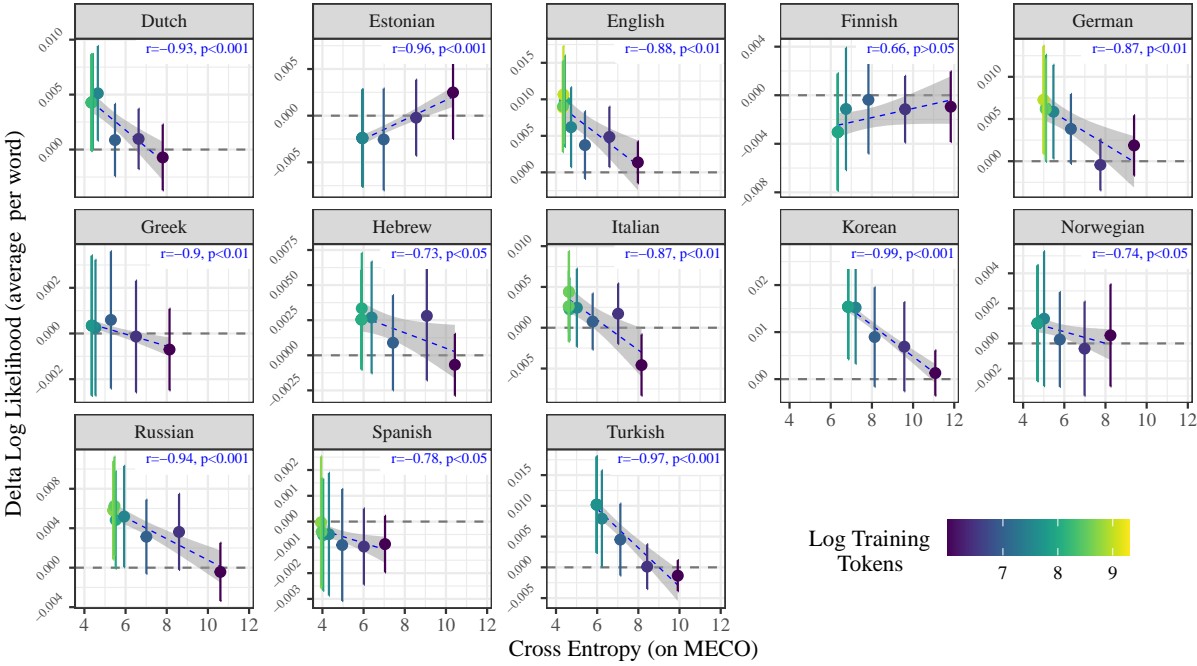

Figure 4: **Results for Entropy Only**: Target models include Rényi entropy with $\alpha = 0.5$ as an additional predictor. We find a negative correlation in 11 out of 13 languages tested.