# OpenReview forum: "Language Model Quality Correlates with Psychometric Predictive Power in Multiple Languages"
_EMNLP/2023/Conference — EMNLP 2023 Main_

### Official Review · Reviewer_fGt4 · 2023-07-20

**Soundness:** 5

**Excitement:**

5: Transformative: This paper is likely to change its subfield or computational linguistics broadly. It should be considered for a best paper award. This paper changes the current understanding of some phenomenon, shows a widely held practice to be erroneous in someway, enables a promising direction of research for a (broad or narrow) topic, or creates an exciting new technique.

**Missing References:**

There are a couple of very recent papers that tackle the same exact problem you are addressing, on the same dataset. They are very recent, so it is of course reasonable that you did not include them in your previous draft, but they should definitely be cited in the amended version of the article.

[1] Wilcox et al. (2023) "Testing the Predictions of Surprisal Theory in 11 Languages." https://arxiv.org/abs/2307.03667

    - In particular §4.3, Figure 4. They found a negative correlation (although non-significant) between perplexity and DeltaLogLik across languages.

[2] de Varda & Marelli (2023) "Scaling in Cognitive Modelling: a Multilingual Approach to Human Reading Times." https://aclanthology.org/2023.acl-short.14/

    - They use XGLM models of different size, and found a negative relationship between parameter size and DeltaLogLik for early fixation measurements, and more nouanced results for other fixation measurements.

Another citation that I think is relevant is [3] which I think nicely complements Hao et al. (2020) (see ll 148 - 153). Not only language models that correlate more strongly with cloze estimates have stronger psychometric predictive power, but training a language model to mirror the empirical human cloze distribution while auto-regressing leads to language models that have higher psychological accuracy.

[3] Eisape, Zaslavsky, & Levy (2020) "Cloze Distillation: Improving Neural Language Models with Human Next-Word Prediction" https://aclanthology.org/2020.conll-1.49/

**Paper Topic And Main Contributions:**

This short paper addresses a current open issue in computational psycholinguistics, namely the relationship between the linguistic (ability to predict the next word) and the psychological accuracy (fit of surprisal / entropy values to human reading time data) of a language model. This problem is addressed by comparing the model fit of surprisal values from different transformer models trained on corpora of varying size on a multilingual eye-tracking dataset. The authors report that better language models are indeed better predictors of human reading data.

**Questions For The Authors:**

Footnote 2: I am not sure I understand your point. If all surprisal values are underestimated (say lower mean and lower variance), then we would have larger ms/bit estimates, but not necessarily worse fit to human reading times. See also work by Merlin & Toneva, 2022 ("Language models and brain alignment: beyond word-level semantics and prediction", https://arxiv.org/pdf/2212.00596.pdf) showing that fine-funing a model on test data increase LM-brain alignment.

**Reasons To Accept:**

[1] The cross-linguistic approach is most welcome! (side note: you might want to take a look at Blasi et al. 2022 "Over-reliance on English hinders cognitive science" (https://www.cell.com/trends/cognitive-sciences/fulltext/S1364-6613(22)00236-4) if you want to make a stronger case on why it is important to have non-English languages in your sample).

[2] I really enjoyed how elegantly the formal proof and the empirical results were combined.

[3] The paper is both technically sound and theoretically rigorous.

[4] The authors train their own transformer models to avoid potential effects of data leakage.

**Reasons To Reject:**

Major:

[1] My main concern regarding this paper is that it considers very small models and reduced training data scales (6 layer transformers with at most 1B training tokens); the problem with this is that the findings in the literature that are in contrast with the QP hypothesis are reported at completely different scales, and I think that everyone would agree that up until GPT-2 (small) larger/lower-perplexity models are better psychometric models. You cite Shain (2023) and Oh & Schuler (2023), but the *smallest* models Oh & Schuler consider are GPT-2 small / GPT-Neo 125M / OPT 125M, and Shain found that GPT-2 was better than GPT-3. So, you are not really testing whether these results hold cross-linguistically without data leakage, because you're testing at a different scale. I see that training multiple larger transformer models can be very demanding and I don't think it is necessary for this kind of study, but the framing should be adjusted and this difference with previous literature should be acknowledged.

Minor:

[1] It would be interesting to see the results for entropy and surprisal tested separately in an appendix. Does the negative relationship between cross-entropy and DeltaLogLik hold when considering surprisal alone? This would facilitate the dialogue with previous research on the relationship between perplexity and DeltaLogLik, which is usually focused on surprisal only.

[2] ll 63-65 "If Surprisal Theory holds, we would expect this power to correlate positively with language models’ quality.". Not necessarily: this is true only (a) if surprisal theory holds, and (b) if your causal graphs on page 3 are correct (i.e., if there isn't any relationship between sθ(w_t) and sH(w_t). As far as I know, surprisal theory per se does not make any strong claim with respect to (b).

[4] ll 289-293 Reporting only p-values without the regression coefficients and standard error is not ideal (also in line 301).

**Reproducibility:**

5: Could easily reproduce the results.

**Reviewer Confidence:**

5: Positive that my evaluation is correct. I read the paper very carefully and I am very familiar with related work.

**Typos Grammar Style And Presentation Improvements:**

[1] I'd say that there is evidence that the probabilistic information that humans deploy in real time is systematically biased with respect to the "true" linguistic distribution. For instance, Goodkind & Bicknell (2021) ("Local word statistics affect reading times independently of surprisal") show that humans are sensitive to local statistics (bigram, trigrams) above and beyond surprisal estimated with language models. This increased sensitivity to local statistics could be interpreted as a systematic deviation of sH(w_t) from s(w_t), which could be reflected in sθ(w_t).

[2] I would at least mention that the dependent variable being fitted are gaze duration values in the main body of the text.

---

> ### Author Rebuttal · Authors · 2023-08-28
>
> Thank you for your review!
>
> > The paper seems motivated as a response to related work showing an inverted, positive relationship between perplexity and quality of fit, but does not test at data sizes (and presumably model sizes) where this inversion was found. My scores for soundness and excitement are based on this assumed motivation and this assumption about models sizes.
>
> See general response (a) above.
>
> > Human sentence processing is usually thought to operate under constraints such as memory limitations which large language models do not operate under. The proof sketch either does not cover the case where some models have such constraints, or does not make it clear how these cases would be covered. For example, if H is a bigram model and theta is an n>2-gram model, then it seems to me H could with enough training have sufficient statistics for all its parameters and converge while theta would improve as n increases, thus theta would seem to provide a poorer fit to r generated from H as theta's number of parameters is increased. It would be nice to have an example like this to explain whether and how the case of memory limitations would be covered.
>
> See general response (c) above.
>
>
> > The paper should specify the model sizes used in the experiments.
>
> We will add this in for camera ready.
>
> Response to Review #3:
>
> Thank you for your review!
>
> > **Major [1]* My main concern regarding this paper is that it considers very small models and reduced training data scales (6 layer transformers with at most 1B training tokens); the problem with this is that the findings in the literature that are in contrast with the QP hypothesis are reported at completely different scales, and I think that everyone would agree that up until GPT-2 (small) larger/lower-perplexity models are better psychometric models. You cite Shain (2023) and Oh & Schuler (2023), but the smallest models Oh & Schuler consider are GPT-2 small / GPT-Neo 125M / OPT 125M, and Shain found that GPT-2 was better than GPT-3. So, you are not really testing whether these results hold cross-linguistically without data leakage, because you're testing at a different scale. I see that training multiple larger transformer models can be very demanding and I don't think it is necessary for this kind of study, but the framing should be adjusted and this difference with previous literature should be acknowledged.
>
> See general response (a) above.
>
> > **Minor [1]** It would be interesting to see the results for entropy and surprisal tested separately in an appendix.
>
> See general response (b) above.
>
> > **Minor [2]** ll 63-65 "If Surprisal Theory holds, we would expect this power to correlate positively with language models’ quality.". Not necessarily: this is true only (a) if surprisal theory holds, and (b) if your causal graphs on page 3 are correct (i.e., if there isn't any relationship between sθ(w_t) and sH(w_t). As far as I know, surprisal theory per se does not make any strong claim with respect to (b).
>
> We agree with this comment. We will clarify the role played by surprisal theory in this claim for CR.
>
> > **Minor [4]** ll 289-293 Reporting only p-values without the regression coefficients and standard error is not ideal (also in line 301).
>
> The p-values are reported from linear regression models, testing the effect of cross entropy on delta log likelihood. For CR we are happy to report the estimate of the cross entropy effect, as well as the coefficient from a correlation analysis.
>
> > **Questions Footnote 2:** I am not sure I understand your point. If all surprisal values are underestimated (say lower mean and lower variance), then we would have larger ms/bit estimates, but not necessarily worse fit to human reading times.
>
> Thank you for this question. To explain our reasoning, there are a few ways in which a model trained on leaked data could provide a poorer fit to human reading times: (i) First, models could likely underestimate RTs if we fit the linking function (i.e., $f_\phi$ in our paper’s notation) between surprisal values and RTs on non-leaked data (which is unseen and potentially higher surprisal). We agree however that in this case, if we also train the linking function on leaked data then it could be possible to recover a good model RT fit. (ii) Second, the underestimation of surprisal may not be uniform across the board—high surprisal words may be “more underestimated” than lower surprisal words—which might hurt delta log likelihood if we use a linear linking function to predict RTs. (iii) Perhaps most importantly, though, evaluating surprisals on data that wasn’t held-out from training may not just underestimate the surprisals, but also lead to hard-to-predict biases—e.g. if we estimated surprisals with an ngram model and no smoothing, the errors in estimates should be proportional to how frequent a context is, which would make it harder to learn a reading time regressor $f_\phi$. Regarding Merlin & Toneva (2022), thanks for pointing this work to us. If we understood their paper correctly, however, (and please do correct us if we are wrong) they finetuned their languages models only on a held out portion of their data (Section 2.4, “We achieve this by finetuning the baseline model with the language modeling objective on a portion of the stimulus text that the brain recordings correspond to.”), but still run their brain analysis on words which were not seen during this finetuning process. If we are right, this would be akin to van Schijndel et al.’s (2018, https://aclanthology.org/D18-1499/) adaptive surprisal models, which are finetuned on a sentence before being used to predict the following one. We will add a discussion of this into the CR.
>
> > There are a couple of very recent papers that tackle the same exact problem you are addressing, on the same dataset. They are very recent, so it is of course reasonable that you did not include them in your previous draft, but they should definitely be cited in the amended version of the article.
>
> We will add these in, thanks for referring us to them.
>
> > **Presentation Improvements [1]** I'd say that there is evidence that the probabilistic information that humans deploy in real time is systematically biased with respect to the "true" linguistic distribution. For instance, Goodkind & Bicknell (2021) ("Local word statistics affect reading times independently of surprisal") show that humans are sensitive to local statistics (bigram, trigrams) above and beyond surprisal estimated with language models. This increased sensitivity to local statistics could be interpreted as a systematic deviation of sH(w_t) from s(w_t), which could be reflected in sθ(w_t).
>
> See general response (c) above.

---

### Official Review · Reviewer_6sX4 · 2023-08-05

**Soundness:** 2

**Excitement:**

4: Strong: This paper deepens the understanding of some phenomenon or lowers the barriers to an existing research direction.

**Paper Topic And Main Contributions:**

This paper studies the relationship between surprisal from transformer models trained from scratch and fixation durations from eye-tracking data in multiple languages.  Results show a negative relationship between perplexity and quality of fit.

**Questions For The Authors:**

Does the proof sketch apply to cases like bigram and n-gram models described above?  If so, how?  If not, why not?

**Reasons To Accept:**

Cross-linguistic psycholinguistic experimentation is essential to testing general theories.

I did not notice any issues with the methodology of the experiments.

**Reasons To Reject:**

The paper seems motivated as a response to related work showing an inverted, positive relationship between perplexity and quality of fit, but does not test at data sizes (and presumably model sizes) where this inversion was found.  My scores for soundness and excitement are based on this assumed motivation and this assumption about models sizes.

The paper should specify the model sizes used in the experiments.

I also was not satisfied with the proof sketch in Section 3.1.  Human sentence processing is usually thought to operate under constraints such as memory limitations which large language models do not operate under.  The proof sketch either does not cover the case where some models have such constraints, or does not make it clear how these cases would be covered.  For example, if H is a bigram model and theta is an n>2-gram model, then it seems to me H could with enough training have sufficient statistics for all its parameters and converge while theta would improve as n increases, thus theta would seem to provide a poorer fit to r generated from H as theta's number of parameters is increased.  It would be nice to have an example like this to explain whether and how the case of memory limitations would be covered.


**Reproducibility:**

2: Would be hard pressed to reproduce the results. The contribution depends on data that are simply not available outside the author's institution or consortium; not enough details are provided.

**Reviewer Confidence:**

4: Quite sure. I tried to check the important points carefully. It's unlikely, though conceivable, that I missed something that should affect my ratings.

---

> ### Author Rebuttal · Authors · 2023-08-28
>
> Thank you for your review!
>
> > The paper seems motivated as a response to related work showing an inverted, positive relationship between perplexity and quality of fit, but does not test at data sizes (and presumably model sizes) where this inversion was found. My scores for soundness and excitement are based on this assumed motivation and this assumption about models sizes.
>
> See general response (a) above.
>
> > Human sentence processing is usually thought to operate under constraints such as memory limitations which large language models do not operate under. The proof sketch either does not cover the case where some models have such constraints, or does not make it clear how these cases would be covered. For example, if H is a bigram model and theta is an n>2-gram model, then it seems to me H could with enough training have sufficient statistics for all its parameters and converge while theta would improve as n increases, thus theta would seem to provide a poorer fit to r generated from H as theta's number of parameters is increased. It would be nice to have an example like this to explain whether and how the case of memory limitations would be covered.
>
> See general response (c) above.
>
>
> > The paper should specify the model sizes used in the experiments.
>
> We will add this in for camera ready.

---

### Official Review · Reviewer_Z8nG · 2023-08-06

**Soundness:** 3

**Excitement:**

4: Strong: This paper deepens the understanding of some phenomenon or lowers the barriers to an existing research direction.

**Paper Topic And Main Contributions:**

The main contribution of this paper is to investigate whether the correlation between model quality and psychometric predictive power (QP hypothesis) does hold cross-linguistically.

**Questions For The Authors:**

- A: Does the addition of Renyi entropy break the formal proof because the proof does not assume Renyi entropy?
- B: How are the equations (2, 4, 6) derived?

**Reasons To Accept:**

- To the best of my knowledge, this paper is the first attempt to investigate whether the correlation between model quality and psychometric predictive power (QP hypothesis) does hold cross-linguistically.

**Reasons To Reject:**

- The reason why the correlation does not hold in Japanese, Finnish, and German remains to be discussed.
- The terms of the equation (1) seem to be reversed, where llh of the nested model (baseline only) is usually subtracted from llh of the nesting model (baseline + surprisal), which may in turn break the formal proof.

**Reproducibility:**

1: Could not reproduce the results here no matter how hard they tried.

**Reviewer Confidence:**

5: Positive that my evaluation is correct. I read the paper very carefully and I am very familiar with related work.

**Typos Grammar Style And Presentation Improvements:**

- l.144: It should, instead be -> It should, instead, be
- l.261: each languages -> each language
- l.296: cross linguistically -> cross-linguistically

---

> ### Author Rebuttal · Authors · 2023-08-28
>
> We would like to thank you for your review!
>
> > The reason why the correlation does not hold in Japanese, Finnish, and German remains to be discussed.
>
> We did not have a longer discussion about language differences due to space, and because, as a short paper, investigating these crosslinguistic differences in detail was not our focus. We will, however, add a longer discussion for CR.
>
> > The terms of the equation (1) seem to be reversed.
>
> Thanks for pointing this out. We had been working with negative LLH in an earlier draft of the article, and did not properly update this equation to take into account the reversed sign. We will change this for CR. Inverting these terms both in eq. 1 and 6 does not affect our proof.
>
> > Does the addition of Renyi entropy break the formal proof because the proof does not assume Renyi entropy?
>
> See General Comment (b) above about entropy in the formal argument.
>
> > How are the equations (2, 4, 6) derived?
>
> (Eq2) is the language modeling loss, written out to highlight how it is equivalent to cross-entropy; (Eq4) is an assumption we make, namely, that the language model’s distribution is a noisy version of the true distribution; (Eq6) uses the definition of mutual information in the first line, of the delta log-likelihood on the second, and the equivalence between the entropy and negative log-likelihood (in this case $\mathrm{H}(R | X)$ and $-\mathrm{llh}(f_{\phi} (\boldsymbol{x}))$) that we would get if we could fit an optimal reading time predictor $\mathrm{rt} = f_{\phi} (\boldsymbol{x})$. We will provide clearer explanations for each of these equations with the added space in the CR.

---

### Meta-Review · Area_Chair_khRr · 2023-09-18

**Recommendation:** 5

**Metareview:**

This paper conducts a large multilingual analysis of the relationship between large language model quality and its ability to predict human reading behavior. They study a diverse set of languages and find a correlation between LLM quality and ability to predict reading times (measured in an eye tracking corpus) in most languages. For a short paper, this paper makes a large contribution, and all reviewers agree that the topic is exciting. Reviewer 2 had a concern about the proof sketch; I think the authors addressed this well during the discussion period. The other reviewers flag some smaller issues, which also appear to have been addressed.

---

### Decision · Program_Chairs · 2023-10-07

**Decision:**

Accept-Main

**Comment:**

This paper conducts a large multilingual analysis of the relationship between large language model quality and its ability to predict human reading behavior. They study a diverse set of languages and find a correlation between LLM quality and ability to predict reading times (measured in an eye tracking corpus) in most languages. For a short paper, this paper makes a large contribution, and all reviewers agree that the topic is exciting. Reviewer 2 had a concern about the proof sketch; I think the authors addressed this well during the discussion period. The other reviewers flag some smaller issues, which also appear to have been addressed.